

# Feature-based detection of automated language models: tackling GPT-2, GPT-3 and Grover

Leon Fröhling[1] and Arkaitz Zubiaga[2]

[1] Leibniz Universität Hannover, Hanover, Germany
[2] Queen Mary University of London, London, UK

## ABSTRACT

The recent improvements of language models have drawn much attention to potential cases of use and abuse of automatically generated text. Great effort is put into the development of methods to detect machine generations among human-written text in order to avoid scenarios in which the large-scale generation of text with minimal cost and effort undermines the trust in human interaction and factual information online. While most of the current approaches rely on the availability of expensive language models, we propose a simple feature-based classifier for the detection problem, using carefully crafted features that attempt to model intrinsic differences between human and machine text. Our research contributes to the field in producing a detection method that achieves performance competitive with far more expensive methods, offering an accessible "first line-of-defense" against the abuse of language models. Furthermore, our experiments show that different sampling methods lead to different types of flaws in generated text.

# INTRODUCTION

Recent developments in Natural Language Processing (NLP) research led to a massive leap in capability of language models. The combination of unsupervised pre-training on massive and diverse datasets (*Radford et al., 2019*) and the introduction of the attention-based transformer architecture (*Vaswani et al., 2017*) allowed increasingly complex models to learn representations of language over a context spanning more than just the next few words, thereby effectively replicating the distribution of human language.

These advances already led to a more comprehensive use of language in a great number of research areas and consumer-oriented applications, as for example in the analysis of biomedical literature (*Beltagy, Lo & Cohan, 2019*), the generation of EEG reports (*Biswal et al., 2019*), the development of more advanced chatbots (*Budzianowski & Vulić, 2019*) and the improvement of grammar- and writing-assistance (*Hagiwara et al., 2019*). However, this newly-gained quality of generated language also increased the fear of its potential abuse by malicious actors (*Solaiman et al., 2019*). Abuse scenarios are mostly based on the effectively vanishing costs for the generation of large amounts of text, allowing malicious actors to leverage the effectiveness of high-volume/low-yield operations like spam, phishing or astroturfing (*Solaiman et al., 2019*; *Ferrara et al., 2016*). While

Corresponding author
Leon Fröhling,
froehling@statistik.uni-hannover.de

*Solaiman et al. (2019)* could not find any evidence of their models being used for automated astroturfing attacks, in which review or comment systems are flooded with generated entries promoting a certain sentiment, an example of how easily text generating models might be abused to influence even policy-making can be found in the American Federal Communications Commission's decision on the repeal of net neutrality rules in 2017 (*Selyukh, 2017*). Attempting to consider the public sentiment through an online comment system, it later turned out that millions of the submitted comments, most of them in favour of repealing net neutrality, were fakes (*Fung, 2017*), automatically generated using a template-based generation model. The little sophistication of the generation approach led to many duplicates and highly similar comments in phrasing and syntax (*Kao, 2017*), drawing attention to the issue in the first place. It is, however, easy to see how one of today's State-Of-The-Art (SOTA) language models might have drowned authentic, human opinions and skewed the final decision without being detected. Similar attacks could potentially overwhelm the news with fake news contents (*Belz, 2019*), manipulate the discourse on social media (*Ferrara et al., 2016*) or impersonate others online or in email (*Solaiman et al., 2019*).

The wider implications of an Internet in which every snippet of written word could with equal probability stem from a human being or a language model are the erosion of fundamental concepts like truth, authorship and responsibility (*Belz, 2019*). *Shevlane & Dafoe (2020)* highlight the potential disruption caused by language models through their ability to impersonate humans in an online world where increasing numbers of human interactions and proportions of social life are hosted, be it in social media, online banking or commerce.

In line with the recommendation given in *Lewandowsky et al. (2012)*, one approach of mitigating the damaging effects of language models is to educate the public about the increasing probability of encountering untrustworthy content online, thereby increasing scepticism and avoiding that factually unsustained information enters a person's belief, from where it would be difficult to retract. However, as argued by *Shevlane & Dafoe (2020)*, such a loss of trust in the habitual informational environment is burdensome. This highlights the need for reliable detection systems in order to tell human and machine generated content apart, preventing the rise of an Internet in which generic nonsense and propaganda-like spam campaigns dominate the public discourse. This paper contributes to the research on the automated detection of machine generated text by being the first to apply a feature-based detection approach to the most recent language models and simultaneously proposing a range of features to be used to that end.

Our experiments with samples from different language generating models show that the proposed feature-based detection approach is competitive with far more complex and computationally more restrictive methods. For its ability to generalize well across different sizes of the same language model, we consider the feature-based classifier a potential "first line-of-defense" against future releases of ever bigger generators. Our research confirms the hypothesis that different sampling methods introduce different kinds of flaws into the generated text, and delivers first insights into which characteristics of text might show these differences the most.

## THE DETECTION PROBLEM

We frame the task of detecting automated language models as a binary classification task where a model needs to determine if an input text is produced by a human or by automated means through a language model. The methods for the detection of machine-generated text presented in this paper take a textual input and assess its provenance based only on the properties of the text, without considering its metadata or veracity, as proposed in similar detection problems (*Baly et al., 2018*; *Thorne & Vlachos, 2018*). To prevent the scenario described above, we expect a detection method to fulfil the following three requirements:

1. *Solaiman et al. (2019)* voice concern for a well-considered trade-off between the maximization of a detector's **accuracy** and the **false positives** it produces. False positives in the present detection context, the incorrect labelling of a human-written text as machine-generated, are especially critical by potentially suppressing human opinions. In a large-scale detection system that automatically filters out texts it considers machine-generated, this could effectively block any written contributions of human authors that happen to have a style similar to what the detector considers typical for language models. This might not only potentially be considered unethical or unlawful, but could also further erode public confidence and trust in the written word online. *A practical detection method must therefore be highly accurate to be able to cope with large-scale adversarial attacks, but may not achieve that at the cost of a high false-positive rate.*

2. Another major fear in the current research into detection methods is the perspective of a "cat and mouse" game (*Solaiman et al., 2019*) between generator and detector, where detection methods are hardly **transferable** between different adversarial generators. Any improvement in language models would then create a temporary advantage for the generating side, persisting until the detector catches up by adapting to the new situation through changes in model architecture or fine-tuning. This would imply that the detection problem could never be resolved, but only temporarily patched. Signs of such a situation arising have been reported by *Radford et al. (2019)* and *Zellers et al. (2019)* who observe that detection models struggle with the increasing complexity of the generating model, *Ippolito et al. (2020)* who find that detection models fail to generalize across different decoding methods used in the generation of texts, and *Bakhtin et al. (2019)*, who note that their detection model does not transfer well across different training corpora. *A detection method needs to be as universal as possible, working well for detecting generations from different language models, trained across different domains and decoded using different sampling methods.*

3. *Gehrmann, Strobelt & Rush (2019)* developed their detection method with the intention to be easy to explain to non-experts and cheap to set up. This follows the recent controversy around availability and reproducibility of SOTA language models, which to a large degree differ only in their increasing financial and computational development costs, effectively restricting the **access** to them. The access-restriction can become harmful when defensive detection methods also rely on the access to such language

models. *Shevlane & Dafoe (2020)* mention the difficulty and cost of propagating defensive measures to potentially harmful AI technologies as an important dimension in the assessment of risks associated with them, implying that a solution is desired that can effectively and easily be used by a large number of users. *Given the anticipated broad impact of language models on human interaction online and usability of the Internet, detection methods should be universally available and easy to set up and adapt.*

## RELATED WORK

This research is aimed at broadening the range of existing detection methods beyond the predominant reliance on the availability of language models by proposing a feature-based approach. To design meaningful features, a good understanding of the properties and limitations of the language generation process is necessary. The following subsections therefore provide an overview of SOTA language generation methods and their limitations, before discussing existing detection methods and subsequently introducing the feature-based approach.

### Language generation

The currently predominating models for language generation are based on the transformer architecture introduced by *Vaswani et al. (2017)*. Its big advantage over previous language models is the more structured memory for long-term dependencies. Even though the bidirectional representation of language, learned by models like BERT (*Devlin et al., 2019*), performs better in many downstream benchmark tasks, unidirectional left-to-right models like GPT-2 (*Radford et al., 2019*) are often the first choice for generating more coherent text (*See et al., 2019*). They allow to intuitively generate text by using the preceding context to estimate a probability distribution over the model's vocabulary, which then only needs to be decoded by sampling the next token from it.

Apart from the new architecture, recent language models profit mainly from the training on ever bigger datasets. *Radford et al. (2019)* trained their model on the WebText dataset, a representation of natural language constructed to be as diverse as possible by spanning many different domains and contexts. The approach to train on as much human-written text as possible is described by *Bisk et al. (2020)* as one of the big milestones in NLP, passing from the usage of domain-specific corpora for training to basically using the whole "written world".

Together with the size of the datasets used for training, the whole training paradigm shifted from task-specific architectures and inputs to unstructured pre-training of language models. First introduced at word-level by *Mikolov et al. (2013)*, *Radford et al. (2019)* took this approach to the sentence-level. By processing as many unstructured, unlabelled, multi-domain and even multilingual texts as possible, the idea is that the models not only get a good understanding of language, but also implicitly learn a variety of potential downstream tasks. The feasibility of this approach was recently confirmed by *Brown et al. (2020)*, whose GPT-3 exhibits strong performance on different NLP

benchmarks, even without any form of task-specific fine-tuning but only through natural language interaction.

In order to effectively leverage the information contained in the ever increasing training datasets into improved language generation ability, the language models have to equally grow in size and complexity. GPT-3 therefore has 175B parameters, more than 100 times as many as its predecessor. *See et al. (2019)* consider current language models to already have enough capacity to effectively replicate the distribution of human language.

Even if a language model perfectly learns the distribution of human language, an equally crucial component in language generation is the choice of the decoding method, i.e. how the next token is sampled from the probability distribution generated by the model. *See et al. (2019)* find that flaws in language generation can be traced back to the choice of decoding method, rather than model architecture or insufficient training. The choice of decoding method can be seen as a trade-off between diversity and quality (*Sun, Schuster & Shmatikov, 2020*; *Hashimoto, Zhang & Liang, 2019*), where sampling from the full distribution leads to diverse, but poor-quality text as perceived by humans, while a likelihood-maximizing sampling method generating only from the most probable tokens leads to high-quality text that lacks diversity and is unnaturally repetitive. *Holtzman et al. (2019)* find the problem of sampling from the full distribution in the increased cumulative likelihood of picking an individually highly unlikely token, causing downward-spirals of text quality which are easy to notice for human readers. When trying to avoid this problem by choosing a likelihood-maximization approach for sampling (e.g. top-k, sampling at every step only from the k most likely tokens), they observe repetition feedback loops which the model cannot escape from and outputs that strongly differ from human language by over-relying on high-likelihood words, making it easy for automated detection approaches to pick up on statistical artifacts.

## Detection approaches

*Solaiman et al. (2019)* introduce a simple categorization of different detection approaches based on their reliance on a language model. In the following, the existing approaches are categorized accordingly and briefly discussed along the dimensions introduced above.

The first category of detection approaches are simple classifiers, trained from scratch based on text samples labelled as either human- or machine-generated. They tend to have relatively few parameters and to be easily deployable. An example is the logistic regression (LR) classifier trained on term frequency-inverse document frequency (tf-idf) features, proposed as a detection baseline by *Clark, Radford & Wu (2019)*. *Badaskar, Agarwal & Arora (2008)* trained a feature-based Support Vector Machine (SVM) classifier, using high-level features to approximate a text's empirical, syntactic and semantic characteristics, trying to find textual properties that differed between human and machine text and could thus be used for discrimination between the two types. Their experiments were limited to the now outdated trigram language models. The main advantages of simple classifiers are their low access- and set-up costs. Because they do not rely on the access to an extensively pre-trained or fine-tuned language model, they can be handled even on individual commodity computers. However, they are hard to adapt, requiring entirely new

training on changing corpora. Because of the sparse literature on them, their performance and transferability are not yet clear, but will be investigated in our experiments.

Zero-shot detection approaches from the second category rely on the availability of a language model to replicate the generation process. An example is the second baseline introduced by *Clark, Radford & Wu (2019)*, which uses the total probability of a text as assessed by a language model for detection. *Gehrmann, Strobelt & Rush (2019)* elaborate on this approach by calculating histograms over next-token probabilities as estimated by a language model and then training LR classifiers on them. While not requiring fine-tuning, zero-shot detection approaches need a language model to work, the handling of which is computationally restrictive. Their performance lags far behind the simple tf-idf baseline (*Clark, Radford & Wu, 2019*; *Ippolito et al., 2020*) and their transferability is questionable, given the need for the detection method in this approach to basically "reverse-engineer" the model-dependent generation process to be successful.

The third category uses pre-trained language models explicitly fine-tuned for the detection task. *Solaiman et al. (2019)* and *Zellers et al. (2019)* add a classifier-layer on top of the language model and *Bakhtin et al. (2019)* train a separate, energy-based language model for detection. While being by far the most expensive method in terms of training time and model complexity, and the least accessible for its reliance on a pre-trained and fine-tuned language model, this approach has so far achieved the highest accuracy on the detection task (*Solaiman et al., 2019*; *Zellers et al., 2019*). However, the discussed lack of transferability across model architectures, decoding methods and training corpora has also been observed with fine-tuned models.

## Feature-based text-classification

The feature-based approach to discriminate between human and machine text is grounded on the assumption that there are certain dimensions in which both types differ. Stylometry—the extraction of stylistic features and their use for text-classification—was introduced by *Argamon-Engelson, Koppel & Avneri (1998)*, and has since been successfully employed for tasks as diverse as readability assessment (*Feng et al., 2010*), authorship attribution (*Koppel, Argamon & Shimoni, 2002*) and, more recently, the detection of fake news (*Pérez-Rosas et al., 2018*; *Rubin et al., 2016*). Even though *Schuster et al. (2019)* consider the detection models of *Zellers et al. (2019)* and *Bakhtin et al. (2019)* examples of well-working, feature-based detectors, their input features are mere vector-space representations of text. *Rubin et al. (2016)* hypothesize that high-level features, specifically designed for the classification problem, expand the possibilities of stylometry classifiers and would thus improve their performance. By building on differences between human and machine text, high-level features make the detection transparent and explainable, offering insights into characteristic behaviour of language models (*Badaskar, Agarwal & Arora, 2008*).

## METHODOLOGY

A feature-based detection approach relies on features that discriminate between human and machine text by modelling properties and dimensions in which both types of text

differ. Logical starting points for the creation of such features are therefore the flaws and limitations of language generation methods. In the following subsection, we categorize the known shortcomings and propose features to capture them, before discussing the choice of a detection model architecture.

## Features

Depending on the choice of the decoding method, the flaws in the generated language differ. However, we establish four different categories to organize them. A comprehensive description and further explanation of the features can be found in the corresponding Supplemental Information.

### Lack of syntactic and lexical diversity

*Gehrmann, Strobelt & Rush (2019)* describe that language models fail to use synonyms and references as humans do, but rather stick to the repetition of the same expressions, leading to a lack of syntactic and lexical diversity in machine text. *Zellers et al. (2020)* observe their models confusing the "who-is-who" in story-telling, and failing to use different references for a text's entities to increase diversity. *See et al. (2019)* find that generated texts contain more verbs and pronouns, and fewer nouns, adjectives and proper nouns than human text, indicating a different use of word types.

This behaviour can be approximated by the use of named entities (NE) and the properties of the coreference chains, as introduced by *Feng et al. (2010)*. Compared to a human author who de-references and varies expressions, language models can be expected to use a larger share of unique NEs and to produce shorter and fewer coreference chains with a higher share of NEs. Additional features can be based on the shift in the part-of-speech (POS) distribution between human and machine texts (*Clark, Radford & Wu, 2019*).

As NE-based features, we use the relative distribution over NE-tags, their per-sentence counts and a number of simple count-based features. The coreference features are similar to those of *Feng et al. (2010)*, all based on coreference chains that indicate the different references made to entities throughout a text. As POS-based features, we use the relative distribution of a text's POS-tags, their per-sentence counts as well as a number of features based on the nouns, verbs, adjectives, adverbs and prepositions proposed by *Feng et al. (2010)*. We use the NE-recognizer and POS-tagger provided in the Python *spaCy* (https://spacy.io/) package to find the NE- and POS-tags, as well as the *neuralcoref* (https://github.com/huggingface/neuralcoref) extension to detect coreference clusters.

### Repetitiveness

The problem of over-using frequent words as described by *Holtzman et al. (2019)* can lead to a large degree of repetitiveness and a lack of diversity in machine-generated texts. *Ippolito et al. (2020)* observe that machine-generated language has 80% of its probability mass in the 500 most common words and *Holtzman et al. (2019)* expose the low-variance of the next-token probabilities over a text as assessed by a language model, showing that machine-generated text almost never dips into low-probability zones as human text characteristically does. Another big problem of machine-generated text is its highly parallel

sentence structure (*Gehrmann, Strobelt & Rush, 2019*) and the occasional repetition of whole phrases (*Jiang et al., 2020*).

We try to expose those statistical differences, assumed to be easiest to be picked up by automated detection methods, through the share of stop-words, unique words and words from "top-lists" in a text's total words. We expect a more diverse, human-written text to have a higher share of unique words and a lower share of stop-words and words from "top-lists". We propose to expose the repetitiveness by calculating the $n$-gram overlap of words (lexical repetition) and POS-tags (syntactic repetition) in consecutive sentences. Human text is expected to be less repetitive both in sentence structure and word choice. We introduce the "conjunction overlap" as a measure of the $n$-gram overlap around *and*-conjunctions to make explicit the reported failure of language models of plainly repeating words around those conjunctions.

We use the stop-words defined by the *spaCy* package and take a list with the top 10,000 words (https://github.com/first20hours/google-10000-english) used in English determined by *Google* to calculate the share of a text's words that are in the top 100, top 1,000 and top 10,000 words of that list. The $n$-gram ($n = [1,2,3]$) overlap of consecutive sentences is represented on a document level by histograms (from 0 to 1 in 10 uniform bins) over the share of repeated word and POS-tag $n$-grams in consecutive sentences.

### Lack of coherence

Even with SOTA language models, the most severe problem of machine-generated text remains the lack of coherence, especially over longer sentences and paragraphs (*Holtzman et al., 2019*; *Brown et al., 2020*). Language model generations are therefore often described as surprisingly fluent on the first read, but lacking any coherent thought and logic on closer inspection (*See et al., 2019*). Closely related is the "topic-drift", where language models struggle to focus on a single topic but cover different, often unrelated topics in a single text (*Badaskar, Agarwal & Arora, 2008*). The lack of coherence is especially blatant for generations sampled with likelihood-maximization, which nevertheless remain hardest to detect for automated detectors due to their lack of sampling-artifacts (*Ippolito et al., 2020*).

The coherence of a text might be approximated by the development of its entities, as introduced by *Barzilay & Lapata (2008)* and used for classification by *Badaskar, Agarwal & Arora (2008)*. The entity-grid representation tracks the appearance and grammatical role of entities through the separate sentences of a text. The assumption is that (locally) coherent text exhibits certain regularities, for example the repetitive presence of a text's main entities in important grammatical roles and only sparse occurrences of less important entities in lesser grammatical roles. We use the *neuralcoref* extension to detect coreference clusters and track the appearance of their entities through the text. As a second layer, we implement an identity-based proxy, considering reappearing, identical noun phrases as the same entity. Using the *spaCy* dependency parser, we assign the roles *Subject (S)*, *Object (O)*, *Other (X)* or *Not Present (-)* to the found entities. Based on the resulting entity grid, we obtain the counts of the 16 possible transitions of entities between consecutive sentences and transform them to relative transition frequencies by normalizing with the total number of transitions.

*Badaskar, Agarwal & Arora (2008)* further propose the use of Yule's Q statistic as described in *Eneva, Hoberman & Lita (2001)* to approximate a text's intra-sentence coherence. Based on the available corpora of human- and machine-generated texts, the assumption is that co-appearances of content-words differ between both types. By requiring a minimal distance of five between the content-words forming a co-appearance pair, the focus is shifted to the model's ability to produce coherent output over a medium-range context length. To discriminate between human and machine text, the texts available in the training corpora are used to calculate a correlation measure for the co-occurrence of content-words in texts from the two different sources. We define content-words as the top 5,000 words from the *Google* top 10,000 list, excluding *spaCy* stop-words and sub-word snippets. Given these correlation scores, separate human- and machine-scores can be calculated for every text, indicating the agreement of that text's content-word co-appearances with the different corpora. The Q statistic is the only corpus-based feature, not exclusively reliant on the text itself.

*Badaskar, Agarwal & Arora (2008)* also use the topic redundancy, approximated by the information loss between a text and its truncated form, as a measure of coherence. The assumption is that human-generated text is more redundant, since it coherently treats a single or few topics without drifting from topic to topic. The text is transformed to a sentence-based vocabulary-matrix representation which can in turn be brought to its eigenspace using a Singular Value Decomposition. By replacing the lowest entries of the eigenvalue diagonal-matrix with 0, the reconstructed matrix is a truncated version of the original. By always setting the lowest 25% of entries to 0, we dynamically adapt to differing text-lengths. Given the original and truncated matrix representation, the information loss is calculated as the squared norm of the element-wise difference between the two matrices. We additionally calculate and include the mean, median, min and max of the truncated matrix and the element-wise difference between the full and truncated matrix.

### Lack of purpose

A final, more qualitative limitation of machine-generated text is its lack of purpose and functionality. While for human text function is generally considered as the "*source of meaning*" (*Bisk et al., 2020*), language models naturally do not have human-like needs or desires (*Gehrmann, Strobelt & Rush, 2019*) and their generations must therefore be considered as void of meaning and purpose.

We approximate the purpose of a text by calculating its lexicon-based topicality scores. We expect human text to contain more sentiment-related keywords and thus score higher in these categories, while being more focussed on fewer categories overall, expressing a single message rather than generating purposelessly drifting text. We also take the share of a text's non-generic content words as a measure of its originality, assuming that human text trying to convey a real message has a higher share.

Based on the 194 categories available by default from the Python *empath* (https://github.com/Ejhfast/empath-client) lexicon-package (*Fast, Chen & Bernstein, 2016*) and five tailored categories (representing spatial properties, sentiment, opinion, logic and ethic),

**Table 1 Validation results.** Classifier accuracies on test set. The classifiers have been fine-tuned with regard to their key parameters using a validation set. Data comes from the different GPT-2 models: small (s), small-k40 (s-k), xl (xl) and xl-k40 (xl-k).

| Training data | Test data | | | | | | | |
|---|---|---|---|---|---|---|---|---|
| | s | | xl | | s-k | | xl-k | |
| | Acc. | AUC | Acc. | AUC | Acc. | AUC | Acc. | AUC |
| Logistic regression | 0.822 | 0.908 | 0.707 | 0.787 | 0.811 | 0.890 | 0.750 | 0.823 |
| SVM | 0.847 | n.a. | 0.704 | n.a. | 0.900 | n.a. | 0.821 | n.a. |
| Neural network | 0.885 | 0.958 | 0.760 | 0.841 | 0.923 | 0.972 | 0.847 | 0.929 |
| Random forest | 0.814 | 0.908 | 0.694 | 0.763 | 0.852 | 0.888 | 0.774 | 0.819 |

we calculate the mean, median, min, max and variance of a text's scores over all categories as features. The same statistics are extracted based only on the "active" categories (empath scores > 0). Additionally, the scores of the text in the tailored categories are used as features.

### Other features

The last set consists of more general, potentially helpful features. The *basic features* are simple character-, syllable-, word- and sentence-counts, both in absolute and relative terms. The *readability features* reflect the syntactic complexity, cohesion and sophistication of a text's vocabulary (*Crossley, Allen & McNamara, 2011*). To test a model's ability of structuring and formatting its generations, we calculate the distribution over punctuation marks, their per-sentence counts as well as the number and average length of paragraphs, shown to be successful in detecting fake news (*Rubin et al., 2016*).

## Classifier

The feature-based detection method proposed in this paper can be considered as a special, binary case of the general automated text categorization problem. We thus follow *Yang & Liu (1999)* in the definition of the task as the supervised learning of assigning predefined category labels to texts, based on the likelihood suggested by the training on a set of labelled texts. Given a text and no additional exogenous knowledge, the trained model returns a value between 0 and 1, indicating the evidence that the document belongs to one class or the other. A hard classifier takes this evidence, compares it to a pre-defined threshold and makes the classification decision (*Sebastiani, 2002*). From the range of available classification models, we consider LR, SVM, Neural Networks (NN) and Random Forests, which have often been reported to show similar performances on the text categorization task (*Zhang & Oles, 2001*; *Joachims, 1998*). We use the implementations of the different models available from the *scikit-learn* (https://scikit-learn.org/) package for our validation trials. We focus our following experiments on the evaluation of NN for the proposed detection problem, based on their superior performance in our validation trials (Table 1).

## EXPERIMENTS

We evaluate our feature-based classifier in a variety of settings, testing it across different generation model architectures, training datasets and decoding methods, thereby covering all main potential influences of a detector's performance.

### Dataset

In our experiments, we use publicly available samples of language model generations and try to detect them among the model's training data, which was either scraped from the Internet or more randomly curated from existing corpora, but in any case of human origin. The biggest part of our data comes from the different GPT-2 model versions, published by *Clark, Radford & Wu (2019)*. We use generations from the smallest (117M parameters; s) and largest GPT-2 model (1,542M parameters; xl), sampled both from the full and truncated (top-$k$ = 40) distribution, to test the transferability of our detectors across model sizes and sampling methods. To evaluate the transferability across model architectures, we include generations from the biggest Grover model (*Zellers et al., 2019*) and from Open-AI's most recent GPT-3 model (*Brown et al., 2020*).

We noticed that a significant share of the randomly scraped and unconditionally generated texts turned out to be website menus, error messages, source code or weirdly formatted gibberish. Since we consider the detection of such low-quality generations as neither interesting nor relevant for the limited impact of their potential abuse, we repeat our experiments on a version of the data that was filtered for "detection relevance". We take inspiration from *Raffel et al. (2019)* in the construction of our filters, filtering out samples that show excessive use of punctuation marks, numbers and line-breaks, contain the words *cookie*, *javascript* or curly brackets, or are not considered as being written in English with more than 99% probability as assessed by the Python *langdetect* (https://github.com/Mimino666/langdetect) package. Like *Ippolito et al. (2020)*, we only consider texts that have at least 192 *WordPiece* (*Schuster & Nakajima, 2012*) tokens. The sizes of the resulting datasets are documented in Table 2. We compare the results of our detectors trained and evaluated on the unfiltered dataset to their counterparts trained and evaluated on the filtered dataset. We expect the filtering to decrease the share of texts without meaningful features, thus hypothesizing that our classifiers perform better on the filtered datasets.

### Evaluation

To evaluate the performance of our detection model, we report its accuracy as the share of samples that are classified correctly, as well as the area under curve (AUC) of the receiver operating characteristic curve (ROC), resulting from the construction of different classification thresholds. While the accuracy is often the sole metric reported in the literature, we argue that it should not be the only metric in assessing a detector's quality. Its inability to include a notion of utility of the different types of errors (*Sebastiani, 2002*) is a major drawback, given the potential severity of false positives as discussed above. This is in line with related detection problems, e.g. the bot detection in social media, where a deliberate focus is on the detector's precision to avoid the misclassification of human

**Table 2 Dataset sizes.** The human text datasets used in our experiments were taken from the samples of human text published with the respective language models and resized to match the size of the machine text datasets.

| Model | Dataset full name | Short Name | Full | | | Filtered | | |
|---|---|---|---|---|---|---|---|---|
| | | | Train | Valid | Test | Train | Valid | Test |
| Machine datasets | | | | | | | | |
| GPT2 | Small-117M | s | 250,000 | 5,000 | 5,000 | 185,622 | 3,732 | 3,722 |
| GPT2 | xl-1542M | xl | 250,000 | 5,000 | 5,000 | 193,052 | 3,868 | 3,851 |
| GPT2 | Small-117M-k40 | s-k | 250,000 | 5,000 | 5,000 | 201,236 | 4,062 | 4,082 |
| GPT2 | xl-1542M-k40 | xl-k | 250,000 | 5,000 | 5,000 | 214,202 | 4,312 | 4,243 |
| GPT3 | 175B | GPT3 | 1,604 | 201 | 201 | 886 | 122 | 101 |
| Grover | Grover-Mega | Grover | 8,000 | 1,000 | 1,000 | 7,740 | 964 | 961 |
| Human datasets | | | | | | | | |
| GPT2 | Webtext | | 250,000 | 5,000 | 5,000 | 190,503 | 3,813 | 3,834 |
| GPT3 | GPT3-webtext | | 1,604 | 201 | 201 | 1,235 | 160 | 155 |
| Grover | realNews | | 8,000 | 1,000 | 1,000 | 7,725 | 972 | 976 |

users as machines (*Morstatter et al., 2016*). Another problem is the sensitivity of accuracy to class skew in the data, influencing the evaluation of detectors (*Fawcett, 2006*) and in extreme cases leading to the trivial classifier (*Sebastiani, 2002*) that effectively denies the existence of the minority class and thus fails to tackle the problem. We therefore decided to report the accuracy, allowing for comparison with existing detection approaches, but also provide the AUC of the ROC as a more comprehensive evaluation metric, effectively separating the evaluation of the classifier from skewed data and different error costs (*Fawcett, 2006*) by combining the notions of specificity (share of correctly detected human texts) and sensitivity (share of correctly detected machine texts).

All reported results are calculated on a held-out test set, using the classifier found to be optimal by a grid search over a range of different parameter constellations and evaluated on validation data. Each of the individual classifiers has thus been optimized across a range of different parameter constellations as defined by a parameter grid using the respective classifier's default optimization method provided in the *scikit-learn* package that was used for training. The classifiers were trained for a maximum of 250 iterations or until convergence on validation data was observed. Please refer to the corresponding Supplemental Information for the underlying parameter grid and the resulting optimal parameter constellations.

## RESULTS

The following results are organized along the different data constellations we trained and evaluated our classifiers on.

### Single-dataset classifiers

In the main part of our experiments, we evaluate detectors trained on samples from a single generation model. We evaluate the resulting detectors not only on the language

**Table 3 Single-dataset classifiers.** Accuracy scores of the classifiers evaluated on generations from the different language models. Along the diagonal (bold), training and test data belong to the same language model.

| Training data | Test data | | | | | | | | | | | |
| --- | --- | --- | --- | --- | --- | --- | --- | --- | --- | --- | --- | --- |
| | s | | xl | | s-k | | xl-k | | GPT3 | | Grover | |
| | Acc. | AUC | Acc. | AUC | Acc. | AUC | Acc. | AUC | Acc. | AUC | Acc. | AUC |
| s | **0.897** | **0.964** | 0.728 | 0.838 | 0.487 | 0.302 | 0.471 | 0.290 | 0.475 | 0.474 | 0.479 | 0.454 |
| xl | 0.740 | 0.937 | **0.759** | **0.836** | 0.504 | 0.434 | 0.489 | 0.382 | 0.468 | 0.423 | 0.516 | 0.485 |
| s-k | 0.338 | 0.247 | 0.445 | 0.328 | **0.927** | **0.975** | 0.808 | 0.924 | 0.537 | 0.769 | 0.502 | 0.671 |
| xl-k | 0.292 | 0.223 | 0.382 | 0.322 | 0.908 | 0.967 | **0.858** | **0.932** | 0.535 | 0.545 | 0.503 | 0.514 |
| GPT3 | 0.436 | 0.234 | 0.452 | 0.316 | 0.736 | 0.821 | 0.658 | 0.749 | **0.779** | **0.859** | 0.589 | 0.654 |
| Grover | 0.333 | 0.285 | 0.439 | 0.422 | 0.662 | 0.785 | 0.643 | 0.738 | 0.537 | 0.552 | **0.692** | **0.767** |

model they were specifically trained on, but also try their transferability in detecting generations from other models.

The feature-based classifier performs better for generations from likelihood-maximizing decoding strategies (Table 3; s-k and xl-k vs. s and xl), as do all the approaches tested in the literature so far. Similarly, the detection of machine-generated texts becomes more difficult with increasing model complexity (Table 3; xl and xl-k vs. s and s-k), indicating that bigger models presumably better replicate human texts statistically. This follows from the baseline results of *Clark, Radford & Wu (2019)* and is also implied by the decreasing performance of our feature-based approach. The performance of the detector learned and evaluated on the GPT-3 model is surprisingly good, being even higher than for the GPT-2 xl generations. Given that GPT-3 has more than 100 times as many parameters, we would have expected GPT-3 generations to be more difficult to detect. However, this might partly be due to the decoding choice, with the top-$p$ = 0.85 sampling used for the GPT-3 generations marking a trade-off between the easier to detect top-$k$ sampling and the harder to detect sampling from the full distribution. Similar reasoning applies to the detection of Grover generations (top-$p$ = 0.94 sampling), which our classifier struggles with most. Another reason might be that the detection of fine-tuned generation models, as is the case with the pre-conditioned article-like Grover generations, is generally more difficult (*Clark, Radford & Wu, 2019*).

Table 3 shows acceptable transferability of our classifiers between models with the same architecture and sampling method, but different complexity. It is easier for a detector trained on samples from a bigger generator (xl and xl-k) to detect samples from a smaller generator (s and s-k) than vice versa. There is no transferability between the different sampling methods, confirming the observations by *Holtzman et al. (2019)* that different sampling methods produce different artifacts, making it impossible for a feature-based detector to generalize between them. To rule out the possibility that the lack of transferability is caused by the corpus-based Q features, we repeat the experiments for detectors trained on all but the Q features (Table A1). The transferability across sampling methods remains abysmal, indicating that the feature-based approach is indeed unable to pick out common flaws produced by different sampling methods.

**Table 4 Feature-set classifiers.** Highlighted in bold are the feature-dataset combinations where a feature-set is far better for either the untruncated or top-k sampling for both GPT-2 dataset sizes. The underscored values correspond to the feature-set and dataset combinations the highlighted values are compared against. The features are sorted in decreasing order of their average accuracy across all datasets.

| Feature-set | Training- and test data | | | | | | | | | | | |
|---|---|---|---|---|---|---|---|---|---|---|---|---|
| | s | | xl | | s-k | | xl-k | | GPT3 | | Grover | |
| | Acc. | AUC | Acc. | AUC | Acc. | AUC | Acc. | AUC | Acc. | AUC | Acc. | AUC |
| Syntactic | 0.859 | 0.944 | 0.733 | 0.826 | 0.845 | 0.925 | 0.780 | 0.865 | 0.714 | 0.803 | 0.627 | 0.692 |
| BasicAbs | 0.822 | 0.910 | 0.716 | 0.794 | 0.817 | 0.900 | 0.747 | 0.827 | 0.679 | 0.766 | 0.602 | 0.664 |
| LexicalDiv | 0.792 | 0.879 | 0.678 | 0.751 | **0.821** | 0.901 | **0.756** | 0.832 | 0.654 | 0.667 | 0.618 | 0.667 |
| InfoLoss | 0.806 | 0.890 | 0.681 | 0.753 | 0.756 | 0.842 | 0.720 | 0.800 | 0.679 | 0.733 | 0.598 | 0.648 |
| Readability | 0.796 | 0.877 | 0.693 | 0.758 | 0.798 | 0.874 | 0.730 | 0.801 | 0.592 | 0.659 | 0.560 | 0.611 |
| Repetitiveness | 0.785 | 0.870 | 0.652 | 0.716 | 0.739 | 0.822 | 0.707 | 0.775 | 0.637 | 0.679 | 0.618 | 0.654 |
| BasicRel | 0.792 | 0.864 | 0.692 | 0.743 | 0.798 | 0.875 | 0.730 | 0.805 | 0.520 | 0.597 | 0.587 | 0.624 |
| NE | **0.795** | 0.886 | **0.677** | 0.751 | 0.725 | 0.807 | 0.660 | 0.727 | 0.632 | 0.673 | 0.543 | 0.549 |
| Empath | 0.710 | 0.786 | 0.627 | 0.682 | 0.703 | 0.778 | 0.624 | 0.676 | 0.649 | 0.727 | 0.572 | 0.595 |
| Formatting | 0.696 | 0.768 | 0.611 | 0.660 | 0.705 | 0.780 | 0.640 | 0.698 | 0.567 | 0.626 | 0.586 | 0.630 |
| Coreference | **0.747** | 0.824 | **0.637** | 0.695 | 0.618 | 0.671 | 0.595 | 0.631 | 0.624 | 0.666 | 0.537 | 0.553 |
| EntityGrid | 0.697 | 0.774 | 0.594 | 0.636 | 0.604 | 0.643 | 0.596 | 0.629 | 0.597 | 0.679 | 0.590 | 0.600 |
| Q | 0.577 | 0.711 | 0.554 | 0.594 | **0.664** | 0.879 | **0.625** | 0.765 | 0.587 | 0.637 | 0.501 | 0.618 |

We finally test the performance of our classifiers when trained and evaluated on the texts from the filtered datasets which are potentially more characteristic and richer in features. As expected, our classifiers perform better, gaining between 1 and 3 percentage-points accuracy across the GPT-2 generations (Table A2). However, this does not hold for GPT-3 and Grover, again hinting at better-curated data.

## Feature-set classifiers

To get an idea of which features are truly important for the performance of the feature-based classifiers, we train and evaluate detectors on the individual subsets of features.

From the results in Table 4 it is apparent that the most important feature subsets in terms of their individual performance are the *syntactic*, *lexical diversity* and *basic* features. While the subsets generally have similar performance for the different sampling methods, we observe that the *NE* and *coreference* features are consistently stronger for the untruncated sampling method, and the *lexical diversity* and Q features for the top-*k* sampling. This is in line with the assumption that untruncated sampling is easier to detect based on more qualitative text characteristics such as coherence and consistency, while generations from top-*k* sampling methods can more easily be detected based on statistical properties.

## Multi-dataset classifiers

Simulating a more realistic detection landscape in which different types of language models are used for the generation of texts, we construct datasets that combine generations from different language models. The combined datasets are composed to optimally balance

**Table 5 Multi-dataset classifiers.** Instances where training and test data belong to the same language model are highlighted (bold).

| Training data | Test data | | | | | | | | | |
|---|---|---|---|---|---|---|---|---|---|---|
| | **s** | **xl** | **s-k** | **xl-k** | **GPT3** | **Grover** | **GPT2-un** | **GPT2-k** | **GPT2** | **All** |
| | Accuracies | | | | | | | | | |
| GPT2-un | 0.827 | 0.726 | 0.508 | 0.497 | 0.473 | 0.458 | **0.817** | 0.500 | 0.636 | 0.602 |
| GPT2-k | 0.323 | 0.430 | 0.921 | 0.839 | 0.515 | 0.602 | 0.381 | **0.871** | 0.607 | 0.616 |
| GPT2 | 0.767 | 0.726 | 0.866 | 0.682 | 0.512 | 0.590 | 0.773 | 0.777 | **0.785** | 0.725 |
| All | 0.809 | 0.690 | 0.880 | 0.772 | 0.510 | 0.643 | 0.760 | 0.824 | 0.782 | **0.755** |
| | AUC | | | | | | | | | |
| GPT2-un | 0.940 | 0.834 | 0.410 | 0.398 | 0.470 | 0.517 | **0.897** | 0.401 | 0.590 | 0.560 |
| GPT2-k | 0.216 | 0.320 | 0.969 | 0.920 | 0.530 | 0.512 | 0.273 | **0.942** | 0.592 | 0.625 |
| GPT2 | 0.932 | 0.800 | 0.940 | 0.829 | 0.566 | 0.593 | 0.877 | 0.881 | **0.865** | 0.787 |
| All | 0.907 | 0.754 | 0.940 | 0.863 | 0.586 | 0.685 | 0.837 | 0.900 | 0.859 | **0.824** |

the contributions of the individual data sources. Their exact composition is documented in Table A3.

Table 5 shows that classifiers trained on combined datasets from the same sampling method (GPT2-un and GPT2-k) lead to good results on the respective individual datasets (s, xl and s-k, xl-k) without outperforming the optimized single-dataset classifiers (Table 3). Their transferability is similar to that of the single-dataset classifier trained on the respective, more difficult dataset (xl, xl-k). When training a classifier on all GPT-2 generations (GPT2), it shows relatively good performance across all individual GPT-2 datasets, but breaks down on the xl-k data. This might hint at the possibility that the detector learns sub-detectors for every single data source, rather than obtaining a universal understanding of the difference between human text and GPT-2 generations.

Finally, we train and evaluate a classifier on the combination of all the different data sources, including generations from GPT-3 and Grover (All). The resulting detector, especially when trained on the subset of features that excludes the corpus-based Q features (Table A4), is surprisingly robust and shows decent performance across all generation models. Its strong performance for the GPT-3 and Grover generations—which are under-represented in the multi-dataset classifiers' training data—might be due to the overall increase in training when compared to the single-dataset classifiers. In total, the multi-dataset classifier is trained on much more and more diverse training samples than the respective single-dataset classifiers for GPT-3 and Grover.

## Ensemble classifiers

After observing that our feature-based classifier is more accurate than the tf-idf baseline in detecting texts from untruncated sampling (s and xl, Table 6), while it is the other way around for texts generated with top-$k$ = 40 sampling (s-k and xl-k, Table 6), we construct ensemble classifiers to take advantage of the differing performances. In the *separate (sep.)* ensemble model variant, we take the individually optimized feature-based- and tf-idf-baseline models' probability estimates for a text to be machine-generated as input to a

**Table 6 Ensemble-classifiers.** The size of the tf-idf vectors in the tf-idf baseline is $n = 100k$.

| Classifier | Training- and test data | | | | | | | | | | | |
|---|---|---|---|---|---|---|---|---|---|---|---|---|
| | s | | xl | | s-k | | xl-k | | GPT3 | | Grover | |
| | Acc. | AUC | Acc. | AUC | Acc. | AUC | Acc. | AUC | Acc. | AUC | Acc. | AUC |
| Baselines | | | | | | | | | | | | |
| Feature-baseline | 0.897 | 0.964 | 0.759 | 0.836 | 0.927 | 0.975 | 0.858 | 0.932 | 0.779 | 0.859 | 0.692 | 0.767 |
| tf-idf-baseline | 0.855 | 0.935 | 0.710 | 0.787 | 0.959 | 0.993 | 0.915 | 0.972 | 0.749 | 0.837 | 0.690 | 0.764 |
| Ensembles | | | | | | | | | | | | |
| LR sep. | 0.877 | 0.959 | 0.740 | 0.831 | 0.966 | 0.995 | 0.920 | 0.976 | 0.761 | 0.844 | 0.689 | 0.764 |
| NN sep. | 0.918 | 0.973 | 0.782 | 0.877 | 0.971 | 0.995 | 0.924 | 0.975 | 0.786 | 0.862 | 0.724 | 0.804 |
| LR super | 0.880 | 0.957 | 0.714 | 0.802 | 0.962 | 0.991 | 0.912 | 0.969 | 0.754 | 0.853 | 0.691 | 0.783 |
| NN super | 0.882 | 0.957 | 0.716 | 0.803 | 0.961 | 0.988 | 0.905 | 0.965 | 0.774 | 0.864 | 0.716 | 0.805 |

meta-learner, which in turn produces the final label estimate. In the *super* ensemble model, we use the probability estimates of all the different, optimized feature-set classifiers, as well as the estimate from the tf-idf-baseline model, as input to a meta-learner. For each of the different ensembles, we train a LR and a Neural Network classifier, following the previously introduced grid-search approach in order to approximate the optimal parameter constellation.

The ensemble models, and especially the *NN sep.* variant built on top of the optimized tf-idf-baseline and feature-based model, outperform the individual classifiers and even improve on their best accuracy by at least 1 percentage-point on each dataset (Table 6). This holds, even though the combination of using NN and the high-dimensional tf-idf baseline necessarily implies strong overfitting to the relatively small input dimensionality, a fact which we observe in the classifiers' near-perfect performances on the training data itself. However, since we explicitly optimized our models on independent validation datasets and not on the training data, we confidently ignore that issue.

## Comparison to results in the literature

Comparing the performance of our feature-based detector to results reported in the literature, we see that the RoBERTa models fine-tuned for the detection task by *Solaiman et al. (2019)* show unmatched accuracies across all model sizes and sampling methods. The accuracies of 96.6% on the xl and 99.1% on the xl-k dataset are impressive, with our best ensemble model lagging behind 18 percentage-points in accuracy on the generations from the full distribution (xl; Table 6). However, *Solaiman et al. (2019)* evaluated their detector only on samples with a fixed length of 510 tokens, potentially giving its accuracy a boost compared to the many shorter, thus harder to detect samples in our test data. The results therefore are not directly comparable. *Ippolito et al. (2020)* report detection results for a fine-tuned BERT classifier on generations from the GPT-2 large model (774M parameters) with a sequence length of 192 tokens. They report an accuracy of 79.0% for generations from the full distribution and 88.0% for top-$k$ = 40 samples. The use of one-token-priming for generation makes their results not directly comparable to ours.

However, as stated by the authors, the priming should only negatively affect the accuracy on the top-k generations. Our strongest ensemble model achieves an accuracy of 78.2% on samples from the untruncated GPT-2 xl model, a generation model twice the size of that used in *Ippolito et al. (2020)* and therefore presumably more difficult to detect. Given the unclear effect of restricting the text length to 192 tokens, compared to our data which includes both longer and shorter texts, we consider our feature-based ensemble classifier to be at least competitive with the reported BERT results. Our best ensemble classifier struggles most with the detection of Grover. While only the fine-tuned Grover model of *Zellers et al. (2019)* scores a strong accuracy of 92.0% on the Grover-Mega data, the fine-tuned BERT and GPT-2 detectors perform similar to our classifier, with reported accuracies of 73.1% and 70.1%, respectively. This suggests that the inability of these detectors might less be due to the detection approach but rather be caused by the highly-curated Grover training data, differing strongly from the more diverse Internet text used to train the non-Grover classifiers.

## DISCUSSION AND FUTURE WORK

Our research into the possibility of using feature-based classifiers for the detection of SOTA language models offers not only an understanding of the method's general performance, but also delivers many insights into more general language model detection issues. We observed low transferability between the detectors of different sampling methods, as well as differing performance of the individual feature sets, indicating that the sampling method choice indeed influences the type of flaws a language model produces in its generations. Our experiments with multi-dataset classifiers indicate that it might be impossible to account for these differences in one single classifier, and that a solution might instead be the construction of sub-classifiers for every single dataset and the combination of their outputs using an ensemble approach. We have also shown that our more quality-focussed features work better than the more statistical tf-idf-baseline for the detection of texts generated from the full distribution, and that ensemble detectors which combine these simple approaches can be competitive with more computationally expensive, language-model-based detectors. Given the transferability observed between different generation model sizes with the same sampling method, we are hopeful that our feature-based approach might work as a "first line-of-defense" against potential releases of ever bigger language models of the same architecture, as was the trend with the last GPT models, without the immediate need to extensively retrain the detector. Given that the dataset used for training has been explicitly crafted to be as diverse as possible and therein covers a wide range of places of discourse on the Internet, we feel confident that our trained classifiers might already in their current form help assessing the origin of text online. However, an important strain of future research would be to systematically evaluate the classifiers' performances in more realistic settings like forum discussions, blog posts or wider social media discourse. Since the training data was constructed to mirror wide parts of the Internet, almost necessarily the potential issue of underrepresentation of minorities arises. The question whether our classifiers show consistent performance across different sources of human text is an important ethical question that requires careful

investigation before deployment, with the aim to ensure that no minorities or non-native speaking groups are discriminated against by a classifier that struggles to detect their speech as human as reliably as it does for other groups. Future work into feature-based detection methods might also include the more detailed evaluation of the contribution of individual features to the overall performance of the classifier, with a possible focus on the search for features that increase transferability between the different sampling methods. Similarly, based on the hypotheses formulated during feature development regarding the role of the different features in distinguishing between human and machine text, a deeper investigation of these linguistic differences could inform the future development and improvement of machine generated language.

## APPENDIX

**Table A1 Single-dataset classifiers, no Q.** Instances where training and test data belong to the same language model are highlighted (bold).

| Training data | Test data | | | | | | | | | | | |
|---|---|---|---|---|---|---|---|---|---|---|---|---|
| | s | | xl | | s-k | | xl-k | | GPT3 | | Grover | |
| | Acc. | AUC | Acc. | AUC | Acc. | AUC | Acc. | AUC | Acc. | AUC | Acc. | AUC |
| s | **0.894** | **0.962** | 0.729 | 0.838 | 0.486 | 0.312 | 0.471 | 0.281 | 0.512 | 0.491 | 0.484 | 0.451 |
| xl | 0.867 | 0.957 | **0.777** | **0.864** | 0.443 | 0.311 | 0.427 | 0.289 | 0.410 | 0.415 | 0.462 | 0.449 |
| s-k | 0.492 | 0.275 | 0.486 | 0.335 | **0.917** | **0.972** | 0.800 | 0.903 | 0.617 | 0.775 | 0.574 | 0.732 |
| xl-k | 0.454 | 0.174 | 0.457 | 0.277 | 0.887 | 0.959 | **0.837** | **0.917** | 0.622 | 0.724 | 0.566 | 0.684 |
| GPT3 | 0.445 | 0.266 | 0.458 | 0.350 | 0.703 | 0.791 | 0.624 | 0.705 | **0.739** | **0.828** | 0.585 | 0.629 |
| Grover | 0.386 | 0.265 | 0.444 | 0.404 | 0.705 | 0.755 | 0.675 | 0.719 | 0.537 | 0.526 | **0.683** | **0.760** |

**Table A2 Single-dataset classifiers, filtered.** Instances where training and test data belong to the same language model are highlighted (bold).

| Training data | Test data | | | | | | | | | | | |
|---|---|---|---|---|---|---|---|---|---|---|---|---|
| | s | | xl | | s-k | | xl-k | | GPT3 | | Grover | |
| | Acc. | AUC | Acc. | AUC | Acc. | AUC | Acc. | AUC | Acc. | AUC | Acc. | AUC |
| s | **0.930** | **0.982** | 0.769 | 0.884 | 0.473 | 0.307 | 0.459 | 0.273 | 0.320 | 0.2139 | 0.431 | 0.43 |
| xl | 0.849 | 0.971 | **0.802** | **0.883** | 0.446 | 0.329 | 0.426 | 0.303 | 0.387 | 0.328 | 0.494 | 0.477 |
| s-k | 0.321 | 0.172 | 0.443 | 0.292 | **0.947** | **0.985** | 0.801 | 0.939 | 0.609 | 0.812 | 0.505 | 0.667 |
| xl-k | 0.216 | 0.099 | 0.360 | 0.242 | 0.910 | 0.974 | **0.861** | **0.933** | 0.637 | 0.660 | 0.514 | 0.721 |
| GPT3 | 0.417 | 0.131 | 0.432 | 0.254 | 0.806 | 0.884 | 0.734 | 0.820 | **0.754** | **0.834** | 0.614 | 0.668 |
| Grover | 0.334 | 0.286 | 0.423 | 0.395 | 0.764 | 0.842 | 0.711 | 0.762 | 0.731 | 0.747 | **0.676** | **0.769** |

**Table A3 Multi-dataset compositions.**

| Name | Machine | | | | | | Human | | |
|---|---|---|---|---|---|---|---|---|---|
| | s | xl | s-k | xl-k | GPT3 | Grover | webtext | GPT3-webtext | realNews |
| | Train datasets | | | | | | | | |
| GPT2-un | 125,000 | 125,000 | – | – | – | – | 250,000 | – | – |
| GPT2-k | – | – | 125,000 | 125,000 | – | – | 250,000 | – | – |
| GPT2 | 62,500 | 62,500 | 62,500 | 62,500 | – | – | 250,000 | – | – |
| All | 60,099 | 60,099 | 60,099 | 60,099 | 1,604 | 8,000 | 236,396 | 1,604 | 8,000 |
| | Valid and test datasets | | | | | | | | |
| GTP2-un | 2,500 | 2,500 | – | – | – | – | 5,000 | – | – |
| GPT2-k | – | – | 2,500 | 2,500 | – | – | 5,000 | – | – |
| GPT2 | 1,250 | 1,250 | 1,250 | 1,250 | – | – | 5,000 | – | – |
| All | 950 | 950 | 950 | 949 | 201 | 1,000 | 3,299 | 201 | 1,500 |

**Table A4 Multi-dataset classifiers, no Q.** Instances where training and test data belong to the same language model are highlighted (bold).

| Training data | Test data | | | | | | | | | |
|---|---|---|---|---|---|---|---|---|---|---|
| | s | xl | s-k | xl-k | GPT3 | Grover | GPT2-un | GPT2-k | GPT2 | All |
| | Accuracies | | | | | | | | | |
| GPT2-un | 0.890 | 0.771 | 0.466 | 0.451 | 0.458 | 0.537 | **0.830** | 0.457 | 0.645 | 0.600 |
| GPT2-k | 0.470 | 0.469 | 0.905 | 0.834 | 0.622 | 0.650 | 0.471 | **0.869** | 0.670 | 0.653 |
| GPT2 | 0.846 | 0.718 | 0.862 | 0.784 | 0.580 | 0.598 | 0.781 | 0.823 | **0.805** | 0.744 |
| All | 0.855 | 0.721 | 0.867 | 0.780 | 0.714 | 0.688 | 0.785 | 0.825 | 0.808 | **0.770** |
| | AUC | | | | | | | | | |
| GPT2-un | 0.962 | 0.859 | 0.291 | 0.271 | 0.444 | 0.450 | **0.909** | 0.277 | 0.594 | 0.558 |
| GPT2-k | 0.197 | 0.293 | 0.968 | 0.917 | 0.757 | 0.703 | 0.245 | **0.942** | 0.594 | 0.628 |
| GPT2 | 0.934 | 0.803 | 0.942 | 0.864 | 0.681 | 0.599 | 0.867 | 0.901 | **0.887** | 0.818 |
| All | 0.938 | 0.808 | 0.942 | 0.856 | 0.755 | 0.746 | 0.871 | 0.898 | 0.888 | **0.856** |

## Funding
The authors received no funding for this work.

## Competing Interests
Arkaitz Zubiaga serves as an Academic Editor for PeerJ Computer Science.

## Author Contributions
- Leon Fröhling conceived and designed the experiments, performed the experiments, analyzed the data, performed the computation work, prepared figures and/or tables, authored or reviewed drafts of the paper, and approved the final draft.
- Arkaitz Zubiaga conceived and designed the experiments, prepared figures and/or tables, authored or reviewed drafts of the paper, and approved the final draft.

## Data Availability

Code is available in the Supplemental Files.

Data is available at GitHub:

- https://github.com/openai/gpt-2-output-dataset
- https://github.com/openai/gpt-3
- https://github.com/rowanz/grover/tree/master/generation_examples.

## Supplemental Information

Supplemental information for this article can be found online at http://dx.doi.org/10.7717/peerj-cs.443#supplemental-information.

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
