# Peer review of "Feature-based detection of automated language models: tackling GPT-2, GPT-3 and Grover"

_PeerJ Computer Science, doi:10.7717/peerj-cs.443_

## Round 0.1 · original submission · Minor Revisions

I propose you to make all necessary changes and additions as soon as possible and resubmitted a revised version.

·

Basic reporting

The manuscript is very well written and clear. Even without a background in NLP, it is possible to understand the methods, simulations, and results. The classifier itself is based on plausible assumptions, allows an intuitive interpretation of (most) features, and performs comparably well as more computationally-intensive competitors. Overall, I think the present manuscript does not have any major flaws. There are still several points that should be improved in a revision:

Detection problem (p.2):
A forth goal that is desirable is the applicability of the classifier to text generated by people who may not be native speakers or members of minorities. This is question of the ethics of using AI methods within a social context. It is sufficient to shortly mention this issue somewhere in the main text (maybe in the discussion or after l.112)

Abbreviations:
The authors should check whether all abbreviations are defined when first mentioned (e.g., SVM, GPT, BERT, tf-idf, POS-tag, ...).

Appendix:
The headings of the feature sets could mention to which subsection in the main text they belong (e.g., coherence, repetitiveness, ...)
Tables: The Labels for rows and columns should clearly distinguish between "Test Data" and "Training Data" as opposed to "Classifier" - since the same type of classifier (neural network) is used for all cases (e.g., Table 3). Moreover, columns and rows should not be switched between tables (e.g., Table 9 & 10).

Discussion:
It could be highlighted that the present work has another benefit: it shows directions for future improvements for machine-generated language.


### Minor Issues:

l. 34: Instead of "bad actors", it would be more appropriate to refer to "actors with questionable/immoral/unethical intentions" (or similar)

l. 56: methods of educating the public may also be difficult to implement effectively from a psychological perspective, e.g.: Lewandowsky, S., Ecker, U. K. H., Seifert, C. M., Schwarz, N., & Cook, J. (2012). Misinformation and its correction: Continued influence and successful debiasing. *Psychological Science in the Public Interest*, *13*(3), 106–131. https://doi.org/10.1177/1529100612451018

l.143: "not to overfit" -> is this a typo? I think this should be "not to underfit"

Table 2: Are the human data sets merged for all training and test sets or matched to the corresponding data set of the language-generation method?

l.383-386: it should be clarified that for each point of the grid, the neural network is trained with an optimization algorithm (backwards propagation?). Currently, this reads as if the models were trained with a grid search - but this appears to be only the case for some of the tuning parameters.

Table 6: It was difficult to detect the pattern highlighted in italics. Underscored numbers are better to detect. Moreover, the column order could be adjusted to have the truncated (s,xl) and full-distribution (s-k,xl-k) datasets next to each other (this holds for all tables). This would in general facilitate recognizing the qualitative patterns discussed by the authors.

l.466: how was overfitting detected? if the model performs well, this cannot be too bad?

Experimental design

Feature definition:
As the authors state correctly, a major benefit of the classifier is that the features have a direct intuitive interpretation and can be communicated to lay audiences and practitioners using these methods. To further highlight this fact, the authors could add a few concrete examples of sentences in which some of the less common features of the classifier are illustrated (either in the main text or on the tables in the appendix). For instance, the new "conjunction overlap" (l.252) seems to refer to matches of the form: ""[x y z] and [x y z]". Similarly, specific examples would facilitate the discussion of named entities and coreference chains. I think this would further strengthen the argument that the feature-based classifier has an intuitive interpretation in contrast to competitors. However, this is merely an optional suggestion and not a mandatory requirement for a revision.

Accuracy measures:
I think it is a good idea to use AUC instead of accuracy. The authors could discuss this in l.79-88. Moreover, it might help to discuss the concepts of sensitivity (probability of detecting machine-generated language) and specificity (counter-probability of the false-positive rate) of binary classifications. The AUC is a measure that takes both criteria into account within a single number.

Ensemble methods:
Why are the classifiers combined with the "tf-idf-baseline models"? According to the argument in l.409-413, it seems more appropriate to combine two feature-based classifiers - one trained with a truncated training set (s-k, xl-k) and another one with the full distribution (s, xl). This makes sense as both classifiers outperform the tf-idf models (l.453). Maybe this is what the authors did, but it is currently not clear.

Validity of the findings

Expectations/hypotheses:
The authors state several hypotheses about the expected direction of differences between machine- and human-generated text. For instance, "We expect a more diverse, human-written text to have a higher share of unique words" (l.248) or "We expect human text to contain more sentiment-related keywords" (l.310). The authors could pick up these hypotheses later in the results section and discuss whether their hypotheses hold in the trained classifiers (this would not require additional tables).

Transferability:
The authors discuss transferability across training sets of different language-generation methods. At some place (e.g., Discussion or l.89-101), it is important to also discuss the issue whether the training sets are representative of "real" text. Put differently, it is not clear whether the classifiers trained with these data would work on data from twitter, facebook, etc. Maybe the authors can specify boundary conditions that need to hold for their classifier to be applicable.

Data:
The GPT-2 samples are currently not available (https://storage.googleapis.com/gpt-2/output-dataset/v1/). The authors state correctly that: "These addresses would need to be updated in the code should they ever change."

Reviewer 2 ·

Basic reporting

I consider the text to be a very good entry into the issue of detecting automatically generated text. The authors reflect on related topics (detection of fake news, authorship attribution, etc.), which gives readers a broader context - the methods are common or similar.

Experimental design

As far as I can judge, I do not see any inconsistencies in the text, the analytical part is given clearly and is based on verifiable methodology and data.

Validity of the findings

The results are absolutely credible.

Additional comments

In further research, from my perspective, the article asks questions to observe the repetition (word forms, grammatical forms) in a small and, on the contrary, a large range of text, probably the cohesion of the text in these different scopes is controlled by opposite tendencies. Even if this were not the case, the text turns out to be inspiring questions based on clearly analyzed data.

---

## Round 0.2 · accepted · Accept

I am completely satisfied with the authors' response.

---

## Author Rebuttal · Round 0.2

Dear Editors

We thank the reviewers for their generous comments and instructive feedback on our work and hope to have edited the manuscript as to address their concerns appropriately. In what follows we respond to each of the comments and describe the changes made to address them.

We hope that you will find the manuscript now suitable for publication in PeerJ Computer Science.

Leon Fröhling and Arkaitz Zubiaga
* * *
# Reviewer 1 (DW Heck)

## *Basic reporting:*

*The manuscript is very well written and clear. Even without a background in NLP, it is possible to understand the methods, simulations, and results. The classifier itself is based on plausible assumptions, allows an intuitive interpretation of (most) features, and performs comparably well as more computationally-intensive competitors. Overall, I think the present manuscript does not have any major flaws. There are still several points that should be improved in a revision:*

## *Detection problem (p.2):*

*A forth goal that is desirable is the applicability of the classifier to text generated by people who may not be native speakers or members of minorities. This is question of the ethics of using AI methods within a social context. It is sufficient to shortly mention this issue somewhere in the main text (maybe in the discussion or after l.112)*

Upon your very true suggestion, we added a comment on the potential problem of a classification model that is not able to recognise minority or non-native speech as human with sufficient reliability, and pointed towards the fact that the consistency of our classifiers across different types of human texts remains to be investigated.

## *Abbreviations:*

*The authors should check whether all abbreviations are defined when first mentioned (e.g., SVM, GPT, BERT, tf-idf, POS-tag, ...).*

We made sure to properly introduce the more technical abbreviations (SVM, tf-idf, POS, …), but refrained from doing so for the names of the most prominent language generation models such as GPT, BERT and Grover. Even though they might be acronyms of longer, more descriptive titles, they are more importantly the names these models are generally known by.

## *Appendix:*

*The headings of the feature sets could mention to which subsection in the main text they belong (e.g., coherence, repetitiveness, ...)*

We added the category of flaws that the feature sets might represent to the feature sets' headings in the corresponding appendix tables.

## *Tables:*

*The Labels for rows and columns should clearly distinguish between "Test Data" and "Training Data" as opposed to "Classifier" - since the same type of classifier (neural network) is used for all cases (e.g., Table 3). Moreover, columns and rows should not be switched between tables (e.g., Table 9 & 10).*

We made sure to maintain consistency between the different tables in indicating the data used for training as row headers and the data used for testing as column headers. We furthermore changed the 'Classifier' header to a more precise 'Test Data' header.

Please note that the changes to the tables are not highlighted in the new manuscript version with the tracked changes due to issues of latexdiff with properly processing these tables. We hope that this is no major problem, especially since these changes were pure layout changes.

*Discussion:*

*It could be highlighted that the present work has another benefit: it shows directions for future improvements for machine-generated language.*

We now mention this as a further benefit (and research prospect) in the concluding discussion.

*### Minor Issues:*

*l. 34: Instead of "bad actors", it would be more appropriate to refer to "actors with questionable/immoral/unethical intentions" (or similar)*

In line with e.g. Solaiman et al (as referenced in the manuscript) we now refer to the former 'bad actors' as 'malicious actors'.

*l. 56: methods of educating the public may also be difficult to implement effectively from a psychological perspective, e.g.: Lewandowsky, S., Ecker, U. K. H., Seifert, C. M., Schwarz, N., & Cook, J. (2012). Misinformation and its correction: Continued influence and successful debiasing. \*Psychological Science in the Public Interest\*, \*13\*(3), 106–131. https://doi.org/10.1177/1529100612451018*

A very interesting and thematically relevant read. I hope that our interpretation of seeing this as an actual argument in line with our point that a high awareness in the public is needed in order to avoid information that was in the best case randomly generated by a language model (which would in the worst case be tailored to spread misinformation) to enter the public's belief – from where it once established is difficult to retract – is in line with your understanding.

*l.143: "not to overfit" -> is this a typo? I think this should be "not to underfit"*

Good point, the original formulation was not very clear. Now it should correctly state the relationship between increasing training dataset sizes and increasing model complexity to effectively leverage it.

*Table 2: Are the human data sets merged for all training and test sets or matched to the corresponding data set of the language-generation method?*

We added a better description of how the human datasets correspond to the different datasets that were used for training the different language models.

*l.383-386: it should be clarified that for each point of the grid, the neural network is trained with an optimization algorithm (backwards propagation?). Currently, this reads as if the models were trained with a grid search - but this appears to be only the case for some of the tuning parameters.*

We rephrased and clarified how the optimization of the different classifier classes and the grid-search to find the optimal classifier among them work together.

*Table 6: It was difficult to detect the pattern highlighted in italics. Underscored numbers are better to detect. Moreover, the column order could be adjusted to have the truncated (s,xl) and full-distribution (s-k,xl-k) datasets next to each other (this holds for all tables). This would in general facilitate recognizing the qualitative patterns discussed by the authors.*

We changed italic numbers to underscored numbers and generally changed the order from s – xl – s-k – xl-k in order to facilitate table readability.

*l.466: how was overfitting detected? if the model performs well, this cannot be too bad?*

Overfitting was detected based on the classifiers' performance on the data it was trained on, to which they are strongly overfitted due to their high-complexity and the relatively low dimensionality of the data. However, this is more on a sidenote since the optimisation was done on separate validation sets and the classifiers therefore still chosen to be the best-performing on independent datasets.

## Experimental Design

### Feature definition:

*As the authors state correctly, a major benefit of the classifier is that the features have a direct intuitive interpretation and can be communicated to lay audiences and practitioners using these methods. To further highlight this fact, the authors could add a few concrete examples of sentences in which some of the less common features of the classifier are illustrated (either in the main text or on the tables in the appendix). For instance, the new "conjunction overlap" (l.252) seems to refer to matches of the form: ""[x y z] and [x y z]". Similarly, specific examples would facilitate the discussion of named entities and coreference chains. I think this would further strengthen the argument that the feature-based classifier has an intuitive interpretation in contrast to competitors. However, this is merely an optional suggestion and not a mandatory requirement for a revision.*

Very good point, we included additional explanation and examples for the mentioned *conjunction overlap, coreference chains, empath features* and *entity-grid features* to the corresponding appendix. We feel like these are the least known of the features and hope to provide a better intuition by our explanations and examples. We consider other features to be either well-established or impossible to break-down into a simple example.

### Accuracy measures:

*I think it is a good idea to use AUC instead of accuracy. The authors could discuss this in l.79-88. Moreover, it might help to discuss the concepts of sensitivity (probability of detecting machine-generated language) and specificity (counter-probability of the false-positive rate) of binary classifications. The AUC is a measure that takes both criteria into account within a single number.*

We kept the introduction and discussion of the AUC in the *evaluation* section, separated from the more qualitative discussion of the need to balance accuracy and false positives in the *detection problem* section. We added reference to the specificity and sensitivity.

### Ensemble methods:

*Why are the classifiers combined with the "tf-idf-baseline models"? According to the argument in l.409-413, it seems more appropriate to combine two feature-based classifiers - one trained with a truncated training set (s-k, xl-k) and another one with the full distribution (s, xl). This makes sense as both classifiers outperform the tf-idf models (l.453). Maybe this is what the authors did, but it is currently not clear.*

We feel like our reasoning to combine the tf-idf baseline with the feature-based classifiers optimised on the untruncated samples should stand. Comparing the results of the feature-based classifiers and the results of the tf-idf baseline classifiers in Table 12 shows that our feature-based classifiers perform better for the untruncated samples (s, xl), while the tf-idf baseline performs better for the top-k samples (s-k, xl-k). We thereby hope to cover both top-k and untruncated samples with the combination of the individually optimal classifiers.

## Validity of the Findings

### Expectations/hypotheses:

*The authors state several hypotheses about the expected direction of differences between machine- and human-generated text. For instance, "We expect a more diverse, human-written text to have a higher share of unique words" (l.248) or "We expect human text to contain more sentiment-related keywords" (l.310). The authors could pick up these hypotheses later in the results section and discuss whether their hypotheses hold in the trained classifiers (this would not require additional tables).*

A very good suggestion of a potentially insightful additional analysis. However, since these hypotheses have mainly been used as qualitative motivations during development of the features and lack any more profound validation, we would not be completely confident in assessing the features individual influence on the classification decision without a more solid linguistic basis. We included this as a potential future research question.

*Transferability:*

*The authors discuss transferability across training sets of different language-generation methods. At some place (e.g., Discussion or l.89-101), it is important to also discuss the issue whether the training sets are representative of "real" text. Put differently, it is not clear whether the classifiers trained with these data would work on data from twitter, facebook, etc. Maybe the authors can specify boundary conditions that need to hold for their classifier to be applicable.*

We added our intuition on this question (training data covers wide parts of the internet, therefore decent transferability could be expected), however, transferability to specific social media platforms etc. would also require further investigation.

*Data:*

*The GPT-2 samples are currently not available (https://storage.googleapis.com/gpt-2/output-dataset/v1/). The authors state correctly that: "These addresses would need to be updated in the code should they ever change."*

The data is actually still available under that address. The quoted URL is only the first part of an URL which is then combined with indications of the specific dataset to be downloaded in the code. The webtext test data would for example be available from https://storage.googleapis.com/gpt-2/output-dataset/v1/webtext.test.jsonl.
* * *
# Reviewer 2 (Anonymous)

*Basic reporting*

*I consider the text to be a very good entry into the issue of detecting automatically generated text. The authors reflect on related topics (detection of fake news, authorship attribution, etc.), which gives readers a broader context - the methods are common or similar.*

*Experimental design*

*As far as I can judge, I do not see any inconsistencies in the text, the analytical part is given clearly and is based on verifiable methodology and data.*

*Validity of the findings*

*The results are absolutely credible.*

*Comments for the Author*

*In further research, from my perspective, the article asks questions to observe the repetition (word forms, grammatical forms) in a small and, on the contrary, a large range of text, probably the cohesion*

*of the text in these different scopes is controlled by opposite tendencies. Even if this were not the case, the text turns out to be inspiring questions based on clearly analyzed data.*

Many thanks for your generous feedback. We agree to your comment regarding the potential difference in text cohesion in different scopes of text and hope that this open question is now sufficiently reflected in our indication of future research direcitons.